# Stool Glycoproteomics Signatures of Pre-Cancerous Lesions and Colorectal Cancer

**DOI:** 10.3390/ijms25073722

**Published:** 2024-03-27

**Authors:** Janine Soares, Mariana Eiras, Dylan Ferreira, Daniela A. R. Santos, Marta Relvas-Santos, Beatriz Santos, Martina Gonçalves, Eduardo Ferreira, Renata Vieira, Luís Pedro Afonso, Lúcio Lara Santos, Mário Dinis-Ribeiro, Luís Lima, José Alexandre Ferreira

**Affiliations:** 1Experimental Pathology and Therapeutics Group, Research Center of IPO Porto (CI-IPOP), RISE@CI-IPOP (Health Research Network), Portuguese Oncology Institute of Porto (IPO Porto), Porto Comprehensive Cancer Center Raquel Seruca (Porto.CCC Raquel Seruca), 4200-072 Porto, Portugal; janinepaivasoares@gmail.com (J.S.); mariana.eiras@ipoporto.min-saude.pt (M.E.); i38795@ipoporto.min-saude.pt (D.F.); daniela.r.santos@ipoporto.min-saude.pt (D.A.R.S.); marta.relvas.santos@ipoporto.min-saude.pt (M.R.-S.); beatriz.santos@ipoporto.min-saude.pt (B.S.); matina.goncalves@ipoporto.min-saude.pt (M.G.); i37310@ipoporto.min-saude.pt (E.F.); luis.afonso@ipoporto.min-saude.pt (L.P.A.); lucio.santos@ipoporto.min-saude.pt (L.L.S.); 2Institute of Biomedical Sciences Abel Salazar (ICBAS), University of Porto, 4050-313 Porto, Portugal; 3REQUIMTE-LAQV, Department of Chemistry, University of Aveiro, 3810-193 Aveiro, Portugal; 4Center for Applied Medical Research, University of Navarra, 31008 Pamplona, Spain; 5i3S—Instituto de Investigação e Inovação em Saúde, Universidade do Porto, 4200-135 Porto, Portugal; 6INEB-Instituto Nacional de Engenharia Biomédica, Universidade do Porto, 4200-135 Porto, Portugal; 7Faculty of Medicine (FMUP), University of Porto, 4200-072 Porto, Portugal; mario.ribeiro@ipoporto.min-saude.pt; 8REQUIMTE-LAQV, Department of Chemistry and Biochemistry, Faculty of Sciences, University of Porto, 4169-007 Porto, Portugal; 9Department of Pathology, Portuguese Oncology Institute of Porto, 4200-072 Porto, Portugal; renata.vieira@ipoporto.min-saude.pt; 10FF-I3ID, University Fernando Pessoa, 4249-004 Porto, Portugal; 11GlycoMatters Biotech, 4500-162 Espinho, Portugal; 12Department of Surgical Oncology, Portuguese Oncology Institute of Porto (IPO-Porto), 4200-072 Porto, Portugal; 13Precancerous Lesions and Early Cancer Management Group, Research Center of IPO Porto (CI-IPOP), Rise@CI-IPOP (Health Research Group), Portuguese Institute of Oncology of Porto (IPO Porto), Porto Comprehensive Cancer Center Raquel Seruca (Porto.CCC Raquel Seruca), 4200-072 Porto, Portugal; 14Department of Gastroenterology, Portuguese Oncology Institute of Porto, 4200-072 Porto, Portugal

**Keywords:** cancer glycosylation, stool glycoproteins, glycoproteomics, pre-cancerous lesions, colorectal cancer

## Abstract

Colorectal cancer (CRC) screening relies primarily on stool analysis to identify occult blood. However, its sensitivity for detecting precancerous lesions is limited, requiring the development of new tools to improve CRC screening. Carcinogenesis involves significant alterations in mucosal epithelium glycocalyx that decisively contribute to disease progression. Building on this knowledge, we examined patient series comprehending premalignant lesions, colorectal tumors, and healthy controls for the T-antigen—a short-chain *O*-glycosylation of proteins considered a surrogate marker of malignancy in multiple solid cancers. We found the T-antigen in the secretions of dysplastic lesions as well as in cancer. In CRC, T-antigen expression was associated with the presence of distant metastases. In parallel, we analyzed a broad number of stools from individuals who underwent colonoscopy, which showed high T expressions in high-grade dysplasia and carcinomas. Employing mass spectrometry-based lectin-affinity enrichment, we identified a total of 262 proteins, 67% of which potentially exhibited altered glycosylation patterns associated with cancer and advanced pre-cancerous lesions. Also, we found that the stool (glyco)proteome of pre-cancerous lesions is enriched for protein species involved in key biological processes linked to humoral and innate immune responses. This study offers a thorough analysis of the stool glycoproteome, laying the groundwork for harnessing glycosylation alterations to improve non-invasive cancer detection.

## 1. Introduction

Colorectal cancer (CRC) is the third most common neoplasia and the second leading cause of cancer-related death globally, posing a significant global health concern [1,2,3]. Early detection has been demonstrated to prolong patients’ lives by more than five years, with approximately 90% of patients experiencing positive outcomes from timely interventions [2,4]. However, this percentage drops to 10% when CRC is diagnosed in advanced stages, particularly with distant metastases, underscoring the major role of screening programs [2]. Among the various screening modalities, stool analysis has emerged as a cornerstone approach due to its non-invasive nature and the possibility of detecting occult blood, a potential indicator of CRC or precancerous lesions [5,6]. Stool-based molecular analysis holds the potential for the identification of individuals bearing colon lesions and enables clinicians to intervene promptly, thereby improving patient outcomes. Furthermore, it may serve as a valuable tool for monitoring CRC patients undergoing treatment, aiding in response assessment and disease progression monitoring [2,7]. The most widely adopted method for CRC screening through stool analysis is the Fecal Immunochemical Test (FIT), which detects the presence of occult blood using antibodies specifically targeting human hemoglobin [6]. FIT demonstrates good sensitivity in detecting cancerous lesions (approximately 70%), namely adenocarcinomas. However, its sensitivity for detecting precancerous lesions decreases to 30%, limiting the overall effectiveness of early screening. To detect advanced lesions at higher risk of progressing to cancer remains a major limitation [6,8]. Furthermore, approximately 30% of FIT-positive cases involve individuals without lesions in the colon and rectum, resulting in unnecessary colonoscopies [6]. Notably, colonoscopy is widely used as the gold standard procedure for colorectal lesion identification due to its high sensitivity for detecting and allowing the removal of pre- and cancerous lesions [2,5]. Nevertheless, colonoscopy is an invasive procedure with inherent limitations such as low patient adherence and high medical and economic burden [9,10]. These limitations underscore the need to develop non-invasive methods based on new molecular biomarkers, such as (glyco)proteins, which have the potential to enhance CRC screening [11].

The pathogenesis of CRC involves intricate molecular alterations, including profound changes in the mucosal epithelium glycocalyx, which play a decisive role in disease progression. Particularly, protein *N*- and *O*-glycosylations have been extensively investigated through in-depth mass spectrometry-based glycomics over the past years, revealing significant alterations linked to disease [12]. Specifically, a decrease in the abundance of core 3 *O*-glycans has been observed in inflamed colon and cancer compared to normal colon tissues [12,13]. Consequently, tumors tend to accumulate core 2 *O*-glycans as well as abnormally shorter *O*-glycan signatures such as the simple disaccharide antigen Thomsen–Friedenreich antigen (TF or T-antigen; Gal(β1–3)GalNAc(α1-O-Ser/Thr), also termed core 1 [13,14]. The T-antigen is mostly absent from normal cells but has been frequently observed in tumors, even though its pathological and clinical consequences remain to be fully elucidated [15]. Moreover, the T-antigen has been detected in liver metastases, the most common hematogenic dissemination in CRC [16]. Furthermore, it has been observed in other solid tumors, such as gastric, lung, prostate, and melanoma, which are linked to aggressiveness [17,18,19,20]. However, apart from certain types of mucins (MUC1, MUC4), the identity of the glycoproteins carrying these glycans remains largely unknown [21,22]. This hampers our comprehension of disease contexts involving the T-antigen and our capacity to effectively explore its biomarker potential.

Building on these observations, we hypothesize that glycoproteins carrying this immature *O*-glycan may be shed or secreted into stools and potentially serve as surrogate biomarkers of malignancy. Therefore, the present study aims to characterize the expression of the T-antigen in premalignant lesions and CRC tumors compared to healthy tissues, providing grounds for further investigations in stools. Our investigation extends to characterizing the T-antigen expression in stool samples from CRC screening programs, providing valuable insights into disease-associated changes in the stool (glyco)proteome. This is expected to not only provide insights into the potential of these glycans in the context of non-invasive CRC screening but also pave the way for more in-depth characterization of the yet unexplored potential of the glycocalyx in this context.

## 2. Results

In this work, we hypothesized that the expression of the immature O-glycan T-antigen could hold potential cancer signatures, given its rare expression in healthy human tissues [15]. Furthermore, we explored the presence of these glycoproteins in stools as potential non-invasive signatures of disease.

### 2.1. T-Antigen in Colorectal Tissues

We started by screening the T-antigen using peanut agglutinin (PNA) lectin, which has been shown to specifically target the T-antigen [23,24]. The T-antigen was evaluated in histologically normal colon and rectum tissues, low-grade (n = 5) and high-grade (n = 3) pre-cancerous lesions, and a series of tumor samples representing all stages of the disease. The T-antigen was not detected in the histologically normal colorectal mucosa but was found in 50% of pre-cancerous lesions (60% low-grade and 33% high-grade dysplasia; Figure 1b), mainly low-grade dysplasia. It was also found in approximately 45% of the cancer cases, predominantly in extracellular secretions in gland lumens and/or mucin pools (Figure 1a). Few cases present scattered expression in the cytoplasm of epithelial cells, most likely belonging to precursors of more extended glycosidic chains in the *O*-glycan biosynthesis pathways (Figure 1a). Despite the high expression of the T-antigen in CRC, we did not find associations between positivity and expression level (intensity and/or extension) with the degree of invasion, clinical stage, tumor location, or survival (Table 1). However, we found that a higher percentage of T-positive tumors were metastatic at the time of presentation (Table 1). In summary, we have shown the disease-specific nature of the T-antigen, providing evidence that it could be expressed in both pre-cancerous lesions as well as cancer, being associated with the presence of metastases. The almost exclusive expression of the T-antigen on secretions and very rare detection in epithelial cells suggest possible remodeling of the mucus layer glycocalyx by the microbiome with a potential impact on disease, as suggested by different reports [25,26].

### 2.2. T-Antigen in Stool

The fact that the T-antigen was almost exclusively observed in secretions facing the lumen of the colorectal mucosa led us to hypothesize that it could be detected in stools and serve as a basis to improve non-invasive CRC screening and follow-ups. We addressed this hypothesis in preliminary settings by screening FIT-positive stools from individuals where lesions with low- and/or high-grade dysplasia were found, as well as CRC patients. We found 13% of T-positive cases among low-grade (3/24) and 67% (8/12) among high-grade dysplasia lesions (Figure 2a,b), generally considered precursors of CRC [27]. Additionally, we observed 33% of T-positive stools from CRC patients (Figure 2a,b). Furthermore, the T-antigen was not detected in the stools collected from individuals who underwent colonoscopy and where no lesions were found. Notably, a comparative analysis between tissues and corresponding stool samples showed a 50% match (Figure 2c). Collectively, these findings reinforce the link between the T-antigen and colorectal pathogenesis. Furthermore, it highlights the potential of the T-antigen to identify high-grade dysplasia, which constitutes a significant milestone for the development of CRC, warranting frequent invasive follow-ups.

### 2.3. Proteomics and Glycoproteomics

The disparity in T-antigen expression between tissues and stool samples has prompted deeper investigations into the stool proteome and glycoproteome expressing T-antigens (Figure 3; Appendix A). For this study, we selected T-positive stool samples from a subset of three individuals from each group (presenting low-grade dysplasia, high-grade dysplasia, or CRC), as well as individuals where no lesions were found. These samples were enriched for T-expressing glycoproteins using PNA lectin affinity chromatography. Subsequently, both the glycoprotein fractions and the flow-through, corresponding to the remaining portion of the stool proteome, were analyzed by tandem mass spectrometry. Before enrichment, the samples were reduced, alkylated, and digested with PNGase F to remove *N*-glycans. According to our experience, the removal of *N*-glycans enhances the interaction of glycoproteins with PNA and reduces potential ambiguities in T-glycopeptides MS/MS spectra annotations.

Overall, we analyzed the proteome and glycoproteome of 12 stool samples, identifying a total of 262 proteins (Appendix A), 67% of which potentially carry the T-antigen (Figure 4a and Appendix A). Both the stool proteome and glycoproteome presented group-specific signatures, proteins commonly expressed protein between cancer and pre-cancerous lesions and common to all groups (Figure 4b,c and Appendix A). A higher number of group-specific glycoproteins were identified in cancer samples in relation to other pathological samples (Figure 4c). Nevertheless, particular attention was devoted to a group of 105 glycoproteins (Figure 4c), found in PNA-enriched fraction (Appendix A), consistently observed in all three samples of at least one of the groups (no lesion (NL), low-grade dysplasia (LG), high-grade dysplasia (HG), cancer). Among these, we found group-specific glycoprotein signatures for cancer (IGHG2, KRT6A, IGLC2, C9, CLCA4, ORM1, HPX, ANXA2P2, HP, A2M, LGALS4) as well as dysplasia (low-grade: COL4A2, PLB1, PRSS2; High-grade: COL4A1, NAALADL1, CAT)—Appendix A. The presence of the T-antigen in these glycoproteins was confirmed by tandem mass spectrometry, as highlighted by the MS/MS spectra for cancer-related glycoproteins alpha-1-acid glycoprotein 1 and alpha-2 macroglobulin (Appendix A). Notably, the number of group-specific glycoproteins was higher in cancer compared to pre-cancerous lesions. Moreover, no group-specific glycoproteins could be found in the stools of FIT-positive cases without colorectal histological alterations. However, we also observed 12 glycoproteins common both to cancer and precancerous lesions groups (Figure 4c) and 43 glycoproteins present in all evaluated groups (Figure 4c; Appendix A). Among cancer and precancerous lesions glycoproteins, we further highlight seven glycoproteins found in high-grade dysplasia as well as cancer (IGHG1, CPA2, MYH2, AZU1, PRTN3, CA1, SMPDL3B), holding potential for non-invasive detection of aggressive traits (Figure 4c; Appendix A). Overall, we found that the stool glycoproteome reflects a wide array of glycoproteins characteristic of epithelial cells of the gastrointestinal tract, such as mucins (MUC2, MUC5AC), collagen fragments (COL4A1, COL4A2 COL4A1, COL4A2), and integrins. Multiple digestive enzymes (CPA1, CTRC, IAP, PA) as well as serum proteins (HBB, IGHG1, SERPINC1, A2M) were also identified, as expected due to the presence of bleeding in these stools. The presence of the T-antigen on these glycoproteins, supported by tandem mass spectrometry, suggests various origins besides the colorectal epithelium for the glycoproteins present in stools. This finding provides a plausible explanation for the absence of a direct correlation between the presence of the T-antigen in colorectal tissues and stools (Figure 2c).

Finally, we comprehensively analyzed the stool proteome and glycoproteome in terms of main biological functions (Figure 4d). In general, we found that the stool proteome of pre-malignant lesions and CRC was enriched for protein species involved in the regulation of glucose transport across the cell membrane, antioxidant activity, and immune responses, namely linked to complement activation and humoral responses against bacteria. This suggests some degree of immune responses against the gut microbiome dysregulations, which become more pronounced in malignant conditions. Notably, all these biological functions were residually represented in the control group, which could be explained by a more stable state of commensal relationship between the microbiome and the host. Also, the number of glycoproteins involved in immune responses increased in cancer compared to pre-malignant lesions. By zooming in on the glycoproteome, we can see a representation of a wider number of biological functions. The presence of glycoproteins linked to immune responses was evident and particularly prominent in cancer and precancerous groups, similar to observations in the proteome. In addition to humoral responses, particularly against bacteria, there was also a relevant number of glycoproteins associated with acute inflammatory responses. Furthermore, CRC exhibited significant enrichment for biological functions linked to alternative pathway complement activation that was not observed in controls and pre-cancerous lesions, suggesting a cancer-specific signature. We also note an enrichment of proteins associated with homeostasis, muscle contraction, and digestion in cancer and precancerous groups that were not evident from proteome data. Collectively, our findings support the existence of pre-cancerous and CRC-specific proteomes and glycoproteomes that warrant further investigation towards non-invasive stool-based CRC screening.

## 3. Discussion

Occult blood in stool serves as a crucial initial indicator of possible alterations in the colorectal mucosa, providing a rationale for CRC screening programs and potentially life-saving interventions. However, limitations related to its sensitivity for detecting CRC and the incapacity to accurately identify advanced lesions (i.e., high-grade dysplasia) pose serious limitations for widespread adoption. Addressing these aspects, we have started by demonstrating that the cancer-associated T-antigen is present in secretions of both CRC and pre-cancerous lesions, holding the potential to be released into stools. This hypothesis was later confirmed by lectin blotting and mass spectrometry analysis, providing a new perspective for glycan-based non-invasive detection using a yet unexplored biological sample in this context.

CRC revealed a higher prevalence of the T-antigen, showing a significant association with the presence of metastases. These observations are consistent with previous reports implicating this glycan in disease spread [28,29]. Namely, the T-antigen has been described as promoting metastasis by facilitating cancer cell interaction with soluble galectins, particularly galectin-3 (Gal-3) [30,31]. These interactions have been shown to drive the adhesion of metastatic cells to the endothelium, homotypic cancer cell aggregation at primary attachment sites, the formation of intravascular metastatic deposits in lungs and bones, and the clonogenic survival and growth of metastatic cancer cell lines [30]. Our findings, together with previous reports documenting the presence of the T-antigen in liver metastases [16], which is the most common hematogenous dissemination route in CRC, reinforce the close link between this glycan and CRC dissemination. This supports ongoing efforts to target the T-antigen-Gal-3 axis with inhibitory molecules and suggests the potential for non-invasive follow-ups of disease progression.

Building on these preliminary findings, we engaged in the assessment of the T status in stool, aiming to create a rationale for glycan-based non-invasive CRC screening. Interestingly, while individuals with healthy colorectal mucosa tested negative for the T-antigen, its elevation was particularly notable in the stools of high-grade adenomas (advanced adenomas). These observations provide an opportunity to enhance FIT’s capacity for aggressive lesion detection and improve CRC screening programs’ effectiveness. Early detection of high-grade dysplasia lesions based on novel biomarkers will be essential for reducing disease burden and improving patient outcomes. Screening for the T-antigen may enable a more efficient allocation of healthcare resources to individuals at higher risk of developing CRC, allowing timely diagnostic and therapeutic interventions. Additionally, this advancement could enhance follow-up care for patients with high-grade adenomas after resection, enabling easier and faster identification of relapse and facilitating closer non-invasive monitoring. Moreover, the association between the T-antigen and metastasis suggests the potential for closer and more frequent cancer progression follow-ups. Overall, our preliminary findings support further investment in assessing the T-antigen to improve stool-based identification of advanced precancerous lesions and progression to CRC. To our knowledge, this is the first report showcasing the potential of glycosylation in stool for non-invasively detecting colorectal lesions.

Finally, we employed a targeted glycoproteomics approach previously demonstrated to be effective in identifying cancer biomarkers [23,32] to investigate the glycoproteins carrying the T-antigen in stools. Our findings challenge the notion of a direct link between the detection of the T-antigen in colorectal tissues and stools. Instead, our results suggest a more complex scenario where a diverse array of glycoproteins from various sources may contribute to the presence of this glycan in stools. These sources include glycoproteins shed or secreted from cells lining the gastrointestinal tract, such as CEACAM5, integrins, cadherins, collagen fragments, and typical colon and rectum mucins like MUC2 and MUC5AC. Additionally, enzymes from secretions of the digestive glands and serum proteins from bleeding were also implicated. Notably, the T-antigen is typically associated with cancer, and its presence in serum proteins and even digestive enzymes can be deemed unusual. In fact, this glycan is widely expressed throughout the body but assumes a cryptic sialylated nature, often appearing in the form of sialyl-T or di-sialylated T-antigens. One plausible explanation for the unexpected identification of the T-antigen in these glycoproteins may involve the activity of enteric commensal and pathogenic bacteria present in the gut microbiome, which are recognized for their ability to utilize sialic acids as a nutrient source through the secretion of sialidases [33]. These are challenging concepts that suggest a connection between the presence of the T-antigen in stools and the microbiome associated with aggressive pre-malignant lesions and diseases. Nevertheless, the possibility of a systemic dysregulation of glycosylation caused by cancer cannot be overruled. Overall, more in-depth studies are needed to disclose these potential relationships fully.

Our study, while exploratory in nature, has provided valuable insights into the glycoproteomes present in both stool samples from individuals without lesions and those from individuals at various stages of disease progression. Through comprehensive analysis, we identified common functional patterns across these samples, regardless of disease severity. Particularly, we observed a notable enrichment of glycoproteins in stool samples of pre-malignant lesions and cancer associated with humoral immune responses, including responses targeting bacteria and acute inflammatory responses such as neutrophil-mediated immunity. An intriguing finding was the significant enrichment of glycoproteins linked to the complement alternative pathway in stools derived from tumors. This pathway, capable of activation in the absence of antibodies, offers a swift and efficient mechanism for enhancing immune responses against pathogens [34]. In summary, our findings suggest typical immune responses against lesions and potential responses against the gut microbiome, known to undergo alterations with disease progression. This sheds light on the intricate interplay between the immune system, microbial communities, and disease states within the gastrointestinal tract [35,36,37,38]. Other relevant biological signatures linked to the disease include changes in the levels of digestive enzymes, regulators of tissue homeostasis, and muscle contraction, suggesting organic dysregulation. In addition to these findings, we identified specific glycoproteins uniquely present in cancer samples, suggesting their potential as biomarkers for cancer detection or monitoring. Among these are HPX, C9, and LGALS4. HPX, synthesized in the liver in response to heme release, is elevated in the serum of CRC patients [39,40,41] and other solid tumors [41], indicating a potential pancarcinomic signature. Notably, HPX has also been reported in the stools of pre-cancerous and CRC patients [42,43], reinforcing its potential as a stool biomarker. However, its role in cancer remains controversial, with evidence suggesting both tumor-suppressive [44] and pro-tumorigenic effects [45]. Notably, sialyl-T-carrying glycoproteoforms of this protein have been found elevated in metastatic CRC [39] and ovarian cancer ascites fluids [46], indicating a significant association with metastasis. C9, primarily produced in the liver, has also been reported to increase in the serum of CRC patients [47,48], particularly in metastatic cases. Its role in cancer also remains poorly understood, with some studies suggesting tumor-promoting effects through inflammation and immune evasion, while others propose immune-mediated tumor destruction [49,50]. LGALS4, expressed by gastrointestinal epithelial cells, plays critical roles in cell adhesion, differentiation, and mucosal barrier formation [51]. It has been implicated in the binding and sorting of glycoproteins in the intestinal epithelium, aiding in the formation of the mucosal barrier and facilitating nutrient absorption [51,52]. Overall, LGALS4 plays a critical role in gastrointestinal physiology and may have implications for diseases such as inflammatory bowel disease and cancer [51]. LGALS4 was also found elevated in the serum and colorectal tumors and associated with poor prognosis [53,54]. In addition, we found several glycoproteins also elevated in high-grade adenomas. In this context, we highlight that AZU1 has also been previously observed as elevated in the stools of adenomas and CRC patients [42], as well as PRTN3, found in polyps and CRC tissues [55]. Additionally, elevated serum levels of PRTN3 have recently been described as an independent poor prognostic factor for progression-free survival and poor response to bevacizumab in patients with metastatic CRC [56]. Moreover, IGHG2, HPT, and A2MG identified in stool-related cancer samples, along with catalase detected in advanced pre-cancerous lesions, have been previously reported elevated in both cancerous and advanced non-cancerous lesions [43]. In summary, a significant body of evidence links many of the stool glycoproteins potentially carrying the T-antigen identified in this study (HPX, C9, LGALS4, AZU1, PRTN3, IGHG2, HPT, A2MG, CATA) to high-grade dysplasia lesions and/or CRC, providing grounds for their exploitation in the context of stool-related disease detection and improving colorectal risk assessment.

## 4. Materials and Methods

### 4.1. FFPE Tissues for Immunohistochemistry Studies

A retrospective patient sample set of 40 formalin-fixed paraffin-embedded (FFPE) CRC tissues collected from the institutional pathology biobank was selected for this study. The sample set included samples collected from non-pediatric patients who underwent surgery at the Portuguese Institute of Oncology of Porto (IPO-Porto) from 2005 to 2012 for resection of colorectal (adeno)carcinomas. Non-epithelial tumors were not considered in the selection. 18 female and 22 male patients, aged between 28 and 76 years (mean: 61 ± 11 years), were elected to participate. Moreover, FFPE tissues from individuals submitted to colonoscopy presenting precancerous lesions were also collected (Low-grade dysplasia (n = 5) and High-grade dysplasia (n = 3)). Adjacent histologically normal mucosa tissues were also analyzed whenever represented on the selected tumor sample. All procedures were approved by the institutional Ethics Board (Approval No. 64/2017 and 22/022, dated 16 March 2017 and 10 February 2022, respectively) and were conducted after signed formal patient consent. Table 2 summarizes the clinicopathological data collected from the patients’ medical records.

### 4.2. Patient Stool Sample Set

A prospective study was conducted at the Gastroenterology Department of IPO-Porto, enrolling individuals with FIT-positive results who underwent colonoscopy. Between April 2022 and July 2023, stool samples were freshly collected the day before patients’ bowel preparation for colonoscopy. The collection and storage of fecal samples were adapted from previous publications [57,58]. Briefly, patients were instructed to collect feces in a sealed plastic container and transport them with an ice bag. Samples were codified and then manually homogenized with a spatula and stored at −80 °C until analysis. A total of 60 samples were included in this study. The samples were selected considering the gender and age of the individuals to maintain cohort homogeneity. Samples were divided into different groups, according to the colonoscopy findings and histological characteristics of biopsy tissue collected during colonoscopy: no lesion found in colonoscopy, low-grade dysplasia, high-grade dysplasia, and patients diagnosed with adenocarcinoma. The individuals’ data are summarized in Table 3. The mean age of individuals was 60.5 years (min–max: 49–83 years). The male/female ratio was approximately 50%. The study received approval from the IPO institutional Ethical Committee (Approval number CES 22/022, dated 10 February 2022) and was conducted following international guidelines, whereby all included patients signed the informed consent.

### 4.3. Immunofluorescence for the T-Antigen

The T-antigen was evaluated in FFPE tissues from pre-malignant and CRC lesions. Briefly, FFPE tissues were prepared as previously described by us [59]. After deparaffinization, rehydration, and antigen retrieval, they were incubated with FITC-labeled PNA lectin (1:1000; Vector Laboratories, Newark, CA, USA) for 1 h at room temperature (RT). T24 WT cells were used as negative controls, whereas T-antigen-positive T24 GCNT1 KO cells were used as positive controls [60]. 4′,6′-diamidino-2-phenylindole dihydrochloride (DAPI, 2.3 × 10^−3^ µg/µL; Thermo Scientific, Waltham, MA, USA) was used as nuclear counterstain. All fluorescence images were acquired on a Leica DMI6000 FFW microscope using Las X software (version 3.0) (Leica, Wetzlar, Germany).

### 4.4. Protein Extraction

Proteins were extracted from 100 mg of stool samples using a lysis buffer solution (50 mM Tris at pH 8, 150 mM NaCl, 1% NP40, 0.5% deoxycholate, and 0.1% SDS) to maximize protein yields. Proteins were supplemented with a cocktail of protease inhibitors in order to maintain protein integrity. After incubation for 20 min on ice, samples were subsequently centrifuged at 2000× *g* for 10 min at 4 °C and supernatant was collected. Proteins were further precipitated with six volumes of acetone and incubated overnight (16 h) at −20 °C. Then, the mixture was centrifuged at 14,000× *g* at 4 °C for 15 min, the supernatant was discarded, and the resulting pellet was air-dried and resuspended by pipetting up and down with the lysis buffer solution supplemented with protease and phosphatase inhibitors. To improve the solubility, samples were subjected to ultrasonic bath cycles at 40 °C and ice cycles. SDS-Page was used to control protein integrity. Finally, the samples were stored at −20 °C for downstream application. Protein samples were quantified using the DC Protein Assay kit.

### 4.5. Lectin Blotting

Stool proteins (50 μg) were transferred in duplicates onto a nitrocellulose membrane pore size 0.45 μm (Amersham^TM^Protran^TM^, Cytiva, Marlborough, MA, USA), using a Hybri-slot apparatus (21052-014; Gibco BRL, Life Technologies, Waltham, MA, USA) under a stabilized vacuum of −0.2 bar. Proteins extracted from T-antigen expressingT24 *GCNT1* KO cells were used as positive controls, whereas BSA was used as a negative control. Protein loads were determined by Ponceau S. Nonspecific bindings were blocked with a Carbo-Free Blocking Solution for 1.5 h at RT. Subsequently, the T-antigen expression was evaluated using biotinylated PNA (VerctorLabs, Newark, CA, USA) at a dilution of 1:10,000 in a mixture of 50% CarboFree and 50% PBS 1x for 1 h at RT. Membranes were washed with PBS-T (PSB with 1% Tween 20) and incubated with the VECTASTAIN^®^ Elite ABD-HRP Reagent (1:10; Vector Laboratories, Newark, CA, USA) for 15 min at RT. The Amersham ECL Prime Detection Reagent was used as a developing reagent. Data analysis was performed through Image Lab Software (version 6.0.1) (Bio-Rad, Hercules, CA, USA) in a ChemiDoc XRS (Bio-Rad, CA, USA).

### 4.6. (Glyco)Proteomics

The stool samples were selected based on the positive results in lectin blotting, except for the NL group, which was chosen randomly due to negative T-antigen expression. Stool proteins were denatured at 80 °C for 10 min, reduced with 5 mM dithiothreitol (20 μL) for 1 h at 60 °C, and alkylated with 10 mM iodoacetamide (20 μL) for 30 min at RT. The reaction was quenched with 2.5 mM DTT (20 μL). After that, proteins were *N*-deglycosylated with PNGase F (1 U/10 µg) overnight (16 h) at 37 °C, followed by incubation at 95 °C for 20 min. Then, protein samples were enriched for glycoproteins carrying T-antigen by lectin affinity chromatography with PNA. Eluates and flow-through for complementary proteomic analysis were digested with trypsin overnight (16 h) at 37 °C. Finally, samples acidified with 0.1% TFA were desalted using peptide desalting spin columns following the manufacturer’s protocol. A nanoLC (Vanquish neoUHPLC) was coupled online to a Qexactive mass spectrometer (Thermo Fisher Scientific, Waltham, MA, USA). Eluent A was aqueous formic acid (0.1%), and eluent B was 0.1% formic acid in 80% acetonitrile. Samples were injected into a trapping column (C18 PepMap 100, 5 μm particle size) and separated in the analytical column (EASY-Spray C18 PepMap, 100 Å, 150 mm × 75 μm I.D and 3 μm particle size) at a flow rate of 0.25 μL/min. The column temperature was set at 35 °C. Glycopeptide separation occurred using a linear gradient of 12–46% eluent B over 50 min for T-antigen-enriched fraction and 80 min for proteomic fraction. Column wash and re-equilibration were warranted before the following injection. The mass spectrometer was operated in the positive ion mode, with an *m*/*z* range from 300 to 2000, a spray voltage of 1.9 kV, and a transfer capillary temperature of 275 °C. Q-Exactive Plus settings were a full scan resolution of 140,000, a fragment scan resolution of 17,500, a fragment scan fixed first mass at 110 *m*/*z*, and an isolation window of 4.0 *m*/*z*. Stepped normalized collision energy (nce) was set to 32, 37, and 41% for the T-antigen-enriched fraction, while nce of 30% was used for proteomic flow-through. Mass spectrometry data were processed using the SequestHT search engine and the Percolator algorithm (Proteome Discoverer 3.0, Thermo Fisher Scientific, Waltham, MA, USA) to validate protein identifications. Data were searched against the human proteome from the SwissProt database. Trypsin was selected as the digestion enzyme, considering up to 3 missed cleavage sites, a precursor ion mass tolerance of 10 ppm, and a product ion tolerance of 0.02 Da. Fixed and variable modifications included carbamidomethylcysteine (+57.021 Da) and oxidation of methionine (+15.995 Da), respectively. In addition, modification of serine and threonine with Hex(1)HexNAc(1) (+365.132 Da; core 1/T-antigen) was also allowed. The final protein list was composed of species identified with high confidence.

### 4.7. Bioinformatics

The whole proteome and PNA-enriched glycoproteins from the four groups addressed in this study (healthy, low-grade dysplasia, high-grade dysplasia, CRC) were characterized according to their biological functions using Cytoscape software (version 3.10.1) [61].

### 4.8. Statistical Analysis

Statistical Package for Social Sciences—SPSS for MacOS (version 27; IBM, Chicago, IL, USA) and GraphPad Prism software (version 9; Dotmatics, San Diego, CA, USA) were used to perform statistical analysis and data visualization. Statistical analysis comprised the Chi-square Test for categorical variables. A *p*-value under 0.05 was considered statistically significant.

## 5. Conclusions

This work presents insights into the unexploited potential of cancer-associated glycans and the stool glycoproteome for cancer screening. It demonstrates that stools from more aggressive lesions may exhibit abnormal immature glycan signatures not seen in healthy tissues, laying the foundation for future exploitation in non-invasive cancer diagnostics. Additionally, it provides a characterization of the stool glycoproteome, identifying several molecular targets shared by more aggressive non-cancerous lesions and cancer, which hold promise for future pre-clinical assessment. Nevertheless, we emphasize the exploratory nature of this study, which may limit the robustness of the results. Future efforts should involve screening larger patient cohorts for both the T-antigen and T-expressing glycoproteoforms, ideally in multicenter settings. The analysis of patients with negative FIT results is also required to fully disclose the potential of these biomarkers for non-invasive CRC screening. Finally, this work showcases the unexplored potential of the CRC glycocode for non-invasive disease detection, providing a framework for addressing other potential relevant alterations in glycosylation in this context.

## Figures and Tables

**Figure 1 ijms-25-03722-f001:**
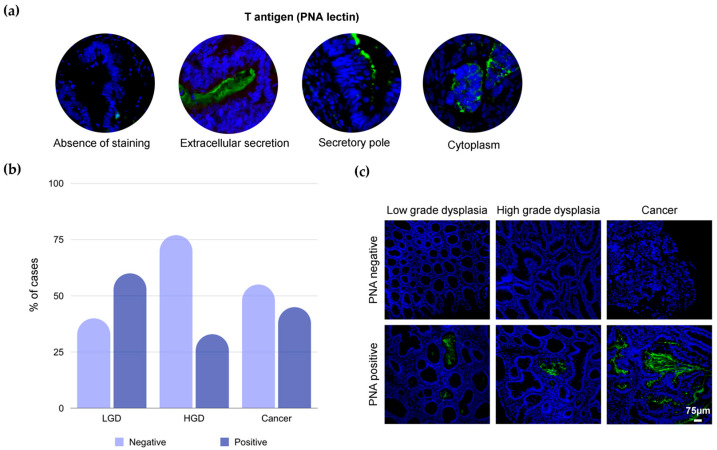
The T-antigen has a mild expression in cancer, as well as in pre-malignant lesions, low-grade dysplasia (LGD), and high-grade dysplasia (HGD), but not in the healthy colon mucosa. (**a**) The T-antigen is expressed in extracellular secretions and the secretory pole and rarely in the cytoplasm of tumors (magnification: 400×). (**b**) The T-antigen is expressed throughout cancer and pre-malignant lesions, although higher expression is present in cancer and LGD. (**c**) T-antigen expression pattern in LGD and HGD and cancer (magnification: 400×).

**Figure 2 ijms-25-03722-f002:**
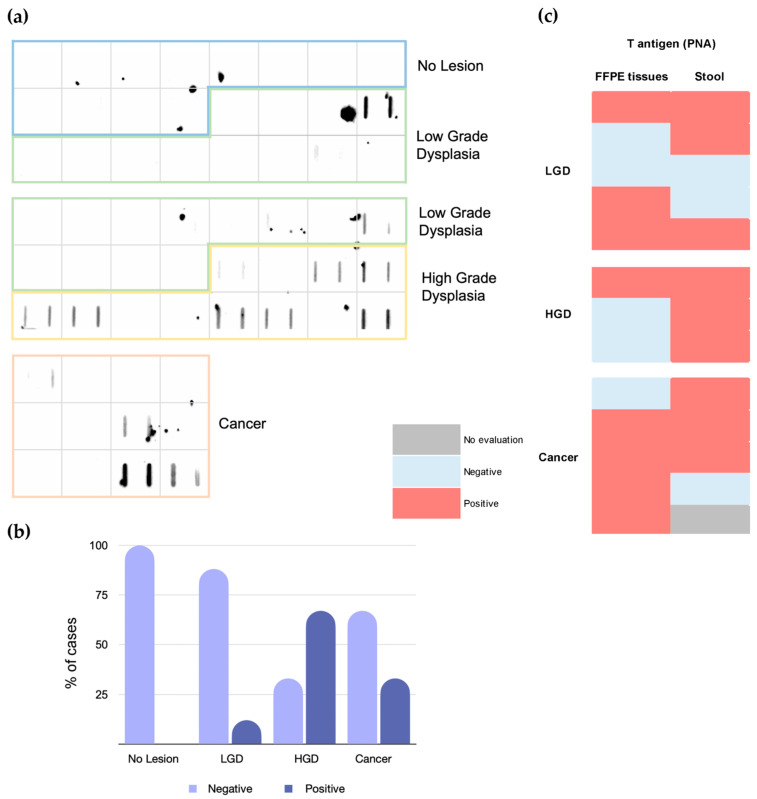
The T-antigen is expressed in the stools of patients with pre-malignant colon lesions (24 cases LGD, 12 cases HGD), in cancer (12 cases), and in no lesion control cases (12), being more frequent in HGD and cancer. (**a**) Silver squares represent each patient, and two biological replicates were used, and (**b**) The T-antigen is more frequent in proteins isolated from the stools of high-grade (67%) and cancer (33% of cancer cases). (**c**) There is a 50% match between T positivity in stools and tissues.

**Figure 3 ijms-25-03722-f003:**
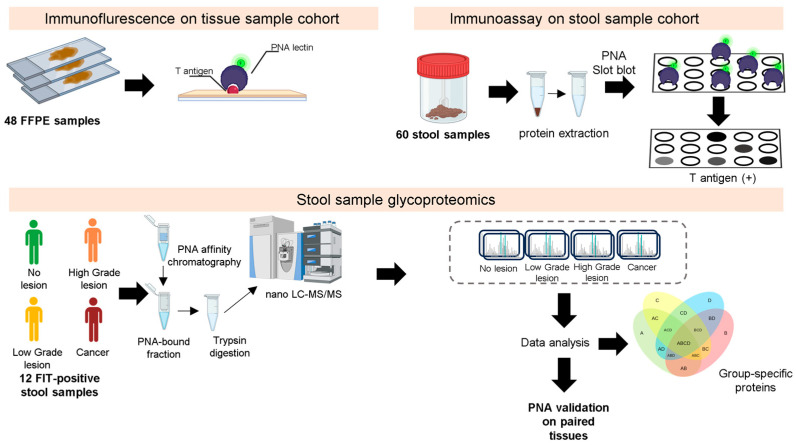
A representative scheme of the proteomics and glycoproteomics workflow was used in this study. Firstly, a cohort of FFPE tissues was screened for the presence of T-antigen, followed by validation in FIT-positive patient stool samples. Briefly, proteins were isolated from stools after homogenization in a lysis buffer solution. The proteins were subsequently reduced, alkylated, and digested with PNGase F to remove *N*-glycans and enriched for T-expressing glycoproteins using PNA lectin affinity chromatography. Both the glycoproteins fraction and unbound proteins were then digested with trypsin and analyzed by nanoLC-MS/MS.

**Figure 4 ijms-25-03722-f004:**
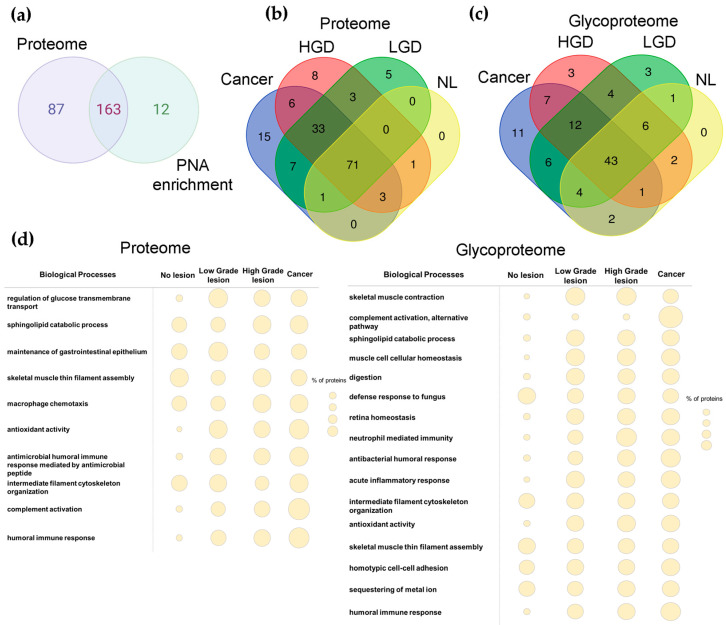
The stool proteomes and glycoproteomes change with disease state (low-grade and high-grade dysplasia and CRC) and present unique proteins that hold the potential for monitoring disease progression. (**a**) A total of 262 proteins were identified, 67% of which were in the glycoproteome fraction, suggesting modification with the T-antigen. (**b**,**c**) The stool proteome and glycoproteome show disease-state-specific glycoproteins. Regarding group specific proteome, there were identified for cancer: HBB, HPX, C7, FGA, LYPD8, TF, LGALS4, PSCA, FOLH1, FGG, C9, TKT, IGHV3-38, ORM2, SERPINB1, HGD: TNNC2, GDPD3, IGKV4-1, PIP, TFF2, CAT, SMR3B, COL4A1, LGD: HLA-DRB5, ATP2A1, CEACAM7, IGLV4-69, CEACAM1, and further highlight six proteins found in LGD and cancer: CALM3, PRTN3, IGHM, CA4, AZU1, CA1. Also, there were identified group-specific glycoprotein signatures for cancer: IGHG2, KRT6A, IGLC2, C9, CLCA4, ORM1, HPX, ANXA2P2, HP, A2M, LGALS4, LGD: COL4A2, PLB1, PRSS2, HGD: COL4A1, NAALADL1, CAT, and we further highlight seven glycoproteins found in HGD and cancer: IGHG1, CPA2, MYH2, AZU1, PRTN3, CA1, SMPDL3B. (**d**) The stool proteome and glycoproteome show disease state-specific functional signatures, by the presence of a higher percentage of proteins in patients bearing lesions and cancer, related to inflammation processes and immune response.

**Table 1 ijms-25-03722-t001:** Association between T-antigen expression and individuals’ clinicopathological characteristics.

	T-Antigen Positive	T-Antigen Negative	
	*n* (%)	*n* (%)	*p*-Value
Stage			
I	1 (5.55)	0 (0)	
II	0 (0)	4 (18.18)	
III	3 (16.67)	7 (31.82)	0.085
IV	14 (77.78)	11 (50.00)	

Tumor (T)			
T1	0 (0)	1 (4.76)	
T2	1 (5.55)	0 (0)	
T3	14 (77.78)	17 (80.95)	0.558
T4	3 (16.67)	3 (14.29)	
Missing information		1 (4.76)	

Lymph node metastases (N)			
N0	5 (27.78)	6 (27.27)	
N1	7 (38.89)	5 (22.73)	
N2	5 (27.78)	9 (40.91)	0.679
N3	1 (5.55)	2 (9.09)	

Distant metastases (M)			
M0	4 (22.22)	12 (54.5)	0.039
M1	14 (77.78)	10 (45.5)	

Tumor location			
Right colon	3 (16.67)	4 (18.19)	
Left colon	13 (72.22)	10 (45.45)	0.152
Rectum	2 (11.11)	8 (36.36)	

**Table 2 ijms-25-03722-t002:** Clinicopathological data associated with tumor tissues were used for this study (n = 40).

	*N* (%)
Stage	
I	1 (2.5)
II	4 (10)
III	10 (25)
IV	25 (65)
Tumor (T)	
T1	1 (2.5)
T2	1 (2.5)
T3	31 (77.5)
T4	6 (15)
Missing Information	1 (2.5)
Lymph node metastases (N)	
N0	11 (27.5)
N1	12 (30)
N2	14 (35)
N3	3 (7.5)
Tumor Location	
Right colon	7 (17.5)
Left colon	23 (57.5)
Rectum	10 (25)

**Table 3 ijms-25-03722-t003:** Demographic data related to stool samples set (n = 60).

Characteristics	No Lesion (n = 12)	Low Grade Dysplasia (n = 24)	High-Grade Dysplasia (n = 12)	Cancer (n = 12)
Age (mean, min–max)	58.0 (50–72)	56.0 (49–74)	63.5 (40–85)	64.5 (51–89)
Gender				
Male	37.5%	37.5%	58.3%	54.2%
Female	62.5%	62.5%	41.7%	45.8%

## Data Availability

The data generated in this study are included in the main article or Appendix A.

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
