# Peer review of "Stool Glycoproteomics Signatures of Pre-Cancerous Lesions and Colorectal Cancer"

_ijms, 2024, doi:10.3390/ijms25073722_

Round 1

Reviewer 1 Report

Comments and Suggestions for Authors

       The objective of this research is to examine the T antigen in colorectal lesions and stools, examining its correlation with the advancement of cancer and detecting modified glycoproteins for enhanced non-invasive cancer identification. Although the study presents a certain scientific interest, there are some concerns, and here are some important comments that will help readers understand the practical significance of these studies:

1.    Establish precise standards for the selection of patients from the institutional pathology biobank. Were there any conditions for inclusion or exclusion?

2.    Describe the methodology used to choose neighboring normal mucosa tissues. Were comparable standards used?

3.    Describe the requirements for a patient to be included in the Portuguese program for the screening of colorectal cancer. Were demographic factors considered?

4.    Describe the procedures for gathering, homogenizing, and storing stool samples. Were established procedures adhered to?

5.    Describe how the results of the colonoscopy will be used to divide the samples. Were blinding techniques applied?

6.    List the sources used to prepare the FFPE tissues. Was it made with this study in mind?

7.    Clearly state why PNA lectin labeled with FITC is used. Were there any procedures involved in validation?

8.    Explain the purpose of the protease inhibitors and lysis buffer. Did protein integrity and yield come into play?

9.    Describe in detail the precipitation and resuspension of proteins. Were quality control procedures followed?

10. Describe the slot-blotting stool protein procedure in detail. Were loading controls applied?

11. Describe the rationale in detail for choosing PNA lectin to analyze T antigens. Were any actions made for validation?

12. Explain why PNGase F and PNA lectin are used. Were other options considered?

13. In Proteome Discoverer 3.0, specify the data processing parameters. Were there any thresholds used?

14.  Use Cytoscape to elaborate on the biological functions that have been described. Are there any particular routes that have been examined?

15.  Clearly explain the advantages of statistical analysis using GraphPad Prism and SPSS. What exams were selected, and why?

16.  Describe the variables that were compared using Chi-square analysis and log-rank tests. Were any adjustments made?

17.  Define selection criteria for tumor samples that encompass all phases of colorectal cancer. Were the factors in balance?

18.  Describe the T antigen detection process in detail. Do any experiments for validation?

19.  Talk about possible confounding variables that could affect the detection of T antigen in stools. In what manner were they addressed?

20.  Deal with sample size constraints, particularly when doing subgroup analyses. Were power calculations done?

21.  Describe the process of choosing samples for proteomic analysis. Were the samples uniform?

22.  Talk about the possible biases caused by glycoprotein enrichment. What was the analysis's approach to them?

23.  Explain the reasoning behind the glycoproteomic analysis's emphasis on the T antigen. Were substitute glycans considered?

24.  Talk about the difficulties in applying research to clinical settings, especially when it comes to CRC screening. What is the validation's next steps?

Comments on the Quality of English Language

-

Author Response

Dear Reviewer,

We thank for the valuable feedback and for the opportunity to improve upon current limitations, which have been comprehensively addressed and implemented, as detailed below:

Comment 1: “Establish precise standards for the selection of patients from the institutional pathology biobank. Were there any conditions for inclusion or exclusion?”

Answer  1: We would like to thank the reviewer for the comment. Addressing this aspect, The section on "FFPE tissues for immunohistochemistry studies" has been updated as follows: "The sample set comprised samples collected from non-pediatric patients undergoing surgery at the Portuguese Institute of Oncology of Porto (IPO-Porto) between 2005 and 2012 for resection of colorectal (adeno)carcinomas. Non-epithelial tumours were not included in the selection."

Comment 2: “Describe the methodology used to choose neighboring normal mucosa tissues. Were comparable standards used?”

Answer  2: We would like to thank the reviewer for the comment. We have clarified the origin of the neighboring normal mucosa: "Adjacent histologically normal mucosa tissues were also analyzed whenever available in the selected tumour samples." The sample selection, including tumour tissue with or without adjacent normal mucosa, was conducted by an experienced pathologist from our pathology department at the time of diagnosis. Their histological analysis enabled the authors to differentiate between normal mucosa and tumour tissues.

Comment 3: “Describe the requirements for a patient to be included in the Portuguese program for the screening of colorectal cancer. Were demographic factors considered?”

Answer  3:  We would like to thank the reviewer’s comment. In the manuscript, in line 447, the authors replaced the text "participating in the Portuguese Colorectal cancer screening program" with "recruited individuals for FIT testing, followed by colonoscopy upon positive results." While most recruited patients participated in the CRC screening program at our hospital, individuals who underwent mucosectomy were also included to increase cases with high-grade dysplasia. Mucosectomy, involving longer anaesthesia, allows removal of larger lesions. Additionally, individuals diagnosed with cancer at IPO Porto's digestive clinic were recruited to increase the number of cancer cases. The authors adjusted the text accordingly, considering demographic factors such as age and gender to ensure group homogeneity.

Comment 4: “Describe the procedures for gathering, homogenizing, and storing stool samples. Were established procedures adhered to?”

Answer  4: Thank you for bringing this to our attention. The procedures for collecting, homogenizing, and storing stool samples have been elaborated upon in the methods section, as requested. These procedures were established based on consensus from various publications, as detailed and referenced in the manuscript.

Comment 5: “Describe how the results of the colonoscopy will be used to divide the samples. Were blinding techniques applied?”

Answer  5: Thank you for the comment. The authors have revised the methods section to incorporate the criteria used to categorize samples into clinical groups. Stratification based on gender and age was employed to ensure group homogeneity. This information has been included in the manuscript (lines 419-420), delineating groups according to colonoscopy findings and histological characteristics: no lesions, low-grade dysplasia, high-grade dysplasia, and adenocarcinoma diagnoses.

Comment 6: “List the sources used to prepare the FFPE tissues. Was it made with this study in mind?”

Answer  6: The FFPE tissue preparation after surgical resection aimed at both pathological characterization and clinical diagnosis. The samples were stored at our hospital’s biobank and made available to the study after ethics approval and specifically for this study. This has been highlighted in the material and methods.

Comment 7: “Clearly state why PNA lectin labeled with FITC is used. Were there any procedures involved in validation?”

Answer  7: PNA lectin labelled with FITC has been widely adopted for detecting the T antigen on glycoengineered cell models, which overexpress the T antigen, through flow cytometry and immunofluorescence. The validation of PNA lectin linkage to the T antigen has been conducted using these cell models, which have been used as positive and negative controls. The reference to these controls has been included in the material and methods section.

Comment 8: “Explain the purpose of the protease inhibitors and lysis buffer. Did protein integrity and yield come into play?”

Answer  8: We thank the reviewer for bringing this to our attention. Regarding protein integrity and yield, the authors extensively tested various lysis buffers and conditions, with and without protease inhibitors, using SDS-PAGE. The protocol was selected based on achieving the best yield and protein integrity. While these results have not been included in the manuscript, they will be addressed in a forthcoming publication focused on protocol details. In addition, protein integrity control regarding potential proteolysis was assessed by SDS-PAGE. Reference to this control was added to the manuscript.

Comment 9: “Describe in detail the precipitation and resuspension of proteins. Were quality control procedures followed?”

Answer  9: We have now described in detail the protocol for precipitation and resuspension of proteins, as requested Regarding the quality control of this procedure, the authors conducted an SDS-PAGE analysis to confirm protein integrity and to identify any signs of proteolysis, ensuring the reliability of the results.

Comment 10: “Describe the slot-blotting stool protein procedure in detail. Were loading controls applied?”

Answer  10: Responding to the reviewer’s concerns, we have now provided a detailed description of the slot blotting protocol. For loading controls, they applied the same amount of stool proteins (50 μg) and confirmed the protein load on the nitrocellulose membrane using Ponceau staining. These aspects have been clarified.

Comment 11: “Describe the rationale in detail for choosing PNA lectin to analyze T antigens. Were any actions made for validation?”

Answer  11: The authors appreciate your comment. PNA lectin was selected to analyze T antigen expression because it specifically targets T antigen (core 1), unlike other commercially available lectins like Artocarpus integrifolia and Maclura pomifera, which also detect core 3. Glycoengineered cell models were utilized to validate PNA lectin's binding to T antigen, as described in prior studies (e.g. Peixoto et al. J Exp Clin Cancer Res 2021).

Comment 12: “Explain why PNGase F and PNA lectin are used. Were other options considered?”

Answer  12: We would like to thank your comment. PNGase F has been chosen because it has been widely employed in the literature to cleave N-linked glycans from various types of oligosaccharides. In this study, PNGase F was utilized to reduce N-glycans, facilitating PNA linkage to T antigen, which might be obscured by extensive glycan chains. Furthermore, the absence of bulky N-glycans significantly facilitates protein annotation from mass spectrometry data. Regarding the PNA lectin, it was selected as the most valuable option to target T antigen in the absence of reliable antibodies or other ligands. These aspects have been now highlighted in the manuscript in the results section.

Comment 13: “In Proteome Discoverer 3.0, specify the data processing parameters. Were there any thresholds used?”

Answer  13: We would like to thank your comment. The authors describe in the method section 4.6. (Glyco)Proteomics, in lines 506 to 514, the specific data processing parameters used in Proteome Discoverer 3.0. The authors also mention in the manuscript the thresholds used for protein identification, in the same section: “precursor ion mass tolerance of 10 ppm, and a product ion tolerance of 0.02 Da”.

Comment 14:Use Cytoscape to elaborate on the biological functions that have been described. Are there any particular routes that have been examined?”

Answer  14. We thank the reviewer for the comment. This was an unbiased approach intended to highlight all major biological functions translated by the (glyco)proteomes at hand, providing a broad overview over the nature of the identified proteoforms.

Comment 15: “Clearly explain the advantages of statistical analysis using GraphPad Prism and SPSS. What exams were selected, and why?”

Answer  15: The authors would like to thank your comment. The authors used the SPSS software for statistical analysis while GraphPad was used for data visualization purposes. This was highlighted in the manuscript in the material and methods section.

Comment 16: “Describe the variables that were compared using Chi-square analysis and log-rank tests. Were any adjustments made?”

Answer  16: We appreciate the reviewer's feedback, which most likely resulted from mentions to Kaplan-Meier and log-rank tests, which were indeed a typographical error for which we apologize. Therefore, references to these tests have been removed. In fact, here only the Chi-square test was utilized to evaluate categorical variables, as indicated in Table 1 of the manuscript. The variables included in the Chi-square analysis are outlined in Table 1 (Stage, Tumour, Lymph Node Metastases, Distant Metastases, and Tumour location compared to T antigen expression). No adjustments were made to the analysis, as the covariates, namely age and gender, were balanced and did not significantly influence the results.

Comment 17: “Define selection criteria for tumour samples that encompass all phases of colorectal cancer. Were the factors in balance?”

Answer  17: We would like to thank the reviewer for the comment. The authors used all tumour samples available at the time, nonetheless the sample cohort is representative of the reality of colorectal cancer disease incidence based on stage. This information was included in the material and methods.

Comment 18: “Describe the T antigen detection process in detail. Do any experiments for validation?”

Answer  18: We thank the reviewer for the comment. The authors describe in the manuscript the process for detecting T antigen, namely in section 4.3. Immunofluorescent for the T antigen (lines 446 to 448) and in section 4.5. Lectin Blotting (lines 473 to 478). These procedures were previously validated in other works of authors' group (e.g. e.g. Peixoto et al. J Exp Clin Cancer Res 2021; Cotton et al. Mol Oncol. 2017). This information was included in the manuscript.

Comment 19: “Talk about possible confounding variables that could affect the detection of T antigen in stools. In what manner were they addressed?”

We thank the reviewer for raising this question. The main aspect that may affect the T antigen is unspecific lectin biding. The amount of protein used in the blots was optimized to avoid this aspect. Also, the blots included positive ( glycoproteins isolated from glycol engineered T24 GCNT1 KO cells overexpressing the T antigen) and negative controls (BSA). This information was included in the material and methods.

Comment 20: “Deal with sample size constraints, particularly when doing subgroup analyses. Were power calculations done?”

Answer  20: We would like to thank the reviewer for the comment. Effectively the power calculation was not applied since the authors used the complete cohort available at time. In the discussion the authors emphasize the explorative nature of the study and the need to increase the number of studied samples.

Comment 21: “Describe the process of choosing samples for proteomic analysis. Were the samples uniform?”

Answer  21: We thank the reviewer for the comment The authors selected samples based on lectin blotting results, specifically targeting cases positive for T antigen expression, except for the "no lesions" group, which tested negative across all parameters. Sample selection was conducted randomly within these criteria. These aspects have been highlighted in the material and methods section.

Comment 22: “Talk about the possible biases caused by glycoprotein enrichment. What was the analysis's approach to them?”

We thank the reviewer for addressing this aspect. Glycoprotein enrichment inherently carries the risk of nonspecific protein binding to columns and, to a lesser extent, lectins. This is typically mitigated by thorough washing before sample incubation and by employing various strategies to isolate disease-specific signatures. In this study, we prioritized protein species absent from control samples known to be negative or express T antigens in amounts bellow the detection threshold. This has been emphasized in the results section.

Comment 23: “Explain the reasoning behind the glycoproteomic analysis's emphasis on the T antigen. Were substitute glycans considered?” 

Answer 23: This work focuses on the T antigen, based on previous reports linking this immature glycan with aggressiveness. This rationale has been presented in the introductory section. The intention was to showcase the potential of exploiting the cancer glycoproteome, building on a significant amount of evidence linking abnormal glycosylation in CRC to aggressiveness. Therefore, we hypothesis that other cancer-associated glycans linked to colorectal cancer may also be an important source of biomarkers and should be explored in future studies. This aspect has been highlighted in the discussion.

Comment 24: “Talk about the difficulties in applying research to clinical settings, especially when it comes to CRC screening. What is the validation's next steps?”

Answer  24: We thank the reviewer for bringing this aspect to our attention. We now discuss this topic on manuscript concluding remarks in the lines 535 to 540 as request.

Reviewer 2 Report

Comments and Suggestions for Authors

The review work " Stool Glycoproteomics Signatures of Pre-Cancerous Lesions 2 and Colorectal Cancer" by Soares and co-researchers examined patient series for understanding premalignant lesions, colorectal tumours, and healthy controls for the T antigen that is considered a substitute marker of malignancy in several solid cancers. The findings of the study seem to be fascinating and interesting, however needs to be improved before acceptance.

·        The abstract still needs modifications specifying a brief significance of solid cancer and relation with biological markers involved.

·        Short sentences should be preferred rather than long continuous sentences.

·        Give some information (brief) related to molecular and clinical significance of T antigens in other solid tumour apart from colorectal cancer.

·        Check for proper referencing throughout manuscript, Eg. “line no 61”.

·        Check for uniformity of writing format throughout manuscript, Eg. “line no. 407-408”.

·        The authors should improve the discussion phrases associated with the findings to the earlier reported studies. Language needs to be modified for better understanding and clarity for readers.

·        The authors have demonstrated their findings using figures/illustrations; however, these are not clear to understand. The quality of figures is necessarily need to be improved.

·        The paper has a lot of elements that can be restructured and put into a conclusion accordingly.

·        The language (English) of the whole manuscript needs to be improved. Simple and uncomplicated sentences can be used in ‘results and discussion section’.  

·        Check for language and clarity for declaration statement’s “mentioned in the last of manuscript”.

Comments on the Quality of English Language

The language (English) of the whole manuscript needs to be improved. Simple and uncomplicated sentences can be used for better understanding.  

Author Response

Dear Reviewer,

We thank for the valuable feedback and for the opportunity to improve upon current limitations, which have been comprehensively addressed and implemented, as detailed below:

Reviewer 2

Comment 1:·”The abstract still needs modifications specifying a brief significance of solid cancer and relation with biological markers involved.”

Answer 1: We thank the reviewer for bringing this aspect to our attention.

Due to abstract size limitations, references to the T antigen in other solid tumours have been incorporated into the introduction, addressing comment 2 accordingly.

Comment 2: “Short sentences should be preferred rather than long continuous sentences.”

Answer 2: The manuscript has been revised accordingly and some sentences have been improved to

respond to this observation.

Comment 3. “Give some information (brief) related to molecular and clinical significance of T antigens in 

other solid tumour apart from colorectal cancer.”

Answer 3: This information is part of the discussion section, namely by addressing the role of this glycan

mediating biding to Gal-3 and its implications for metastasis development.

Comment 4: Check for proper referencing throughout manuscript, Eg. “line no 61”.

We thank the reviewer for the comment. All references ha

Comment 5: Check for uniformity of writing format throughout manuscript, Eg. “line no. 407-408”.

Answer 5: We thank the reviewer for bringing this to our attention. This has been implemented.

Comment 6: “The authors should improve the discussion phrases associated with the findings to the

earlier reported studies. Language needs to be modified for better understanding and clarity for readers.”

Answer 6: We thank the reviewer. This has been implemented as requested.

Comment 7: “The authors have demonstrated their findings using figures/illustrations; however, these

are  not clear to understand. The quality of figures is necessarily need to be improved.”

Answer 7: We apologise for the lack of clarity. This has been improved as requested.

Comment 8: The paper has a lot of elements that can be restructured and put into a conclusion”

Answer 8: The conclusion has been improved as requested while maintaining a clear focus on the main

takeaway.

Comment 9:The language (English) of the whole manuscript needs to be improved. Simple and

uncomplicated sentences can be used in ‘results and discussion section’. “

Answer 9. While we fell that the manuscript is rather easy to fell, attempts have been made to improve

some sentences.

Comment 10: “Check for language and clarity for declaration statement’s “mentioned in the last of

manuscript”.

Answer 10: We thank the reviewer for bringing this aspect to our attention. This has been improved as requested

Reviewer 3 Report

Comments and Suggestions for Authors

In this work, authors Soares et. al. are presenting a study assessing the expression of the T antigen in premalignant lesions and colorectal cancer (CRC) tumors compared to healthy tissues. The work also extends to studying the T antigen expression in CRC stool samples to understand the changes in glycoproteome that can be associated with disease and thus serve as potential biomarkers for its early detection. Overall, the research work is highly significant and the manuscript is well written. However, authors can further improve it by making some minor revisions as discussed below.

Please provide the full form of PNA lectin. The abbreviation first appears in Section 2.1 line 123 without the full form. 

Authors should consider breaking Figure 3 into at least two different figures since it is too crowded in its current form. Text in Figure 3 (e) is hardly legible due to low resolution. Perhaps Figure 3 (b) – (d) that are showing results can be separated from (a) that is showing workflow.

Page 14 Lines 468-469 “After that, proteins were N-deglycosylated with PNGase F (1U/10μg) overnight at 37 °C” Please mention the number of hours for this incubation step. Overnight incubation is a vague term. Same for Trypsin digestion. Please provide the hours of incubation.

Please provide the volumes of 5 mM dithiothreitol and 10 mM iodoacetamide used for reduction, alkylation and quenching? 

Provide the volume and amount of sample injected for LC-MS analysis. 

Author Response

Dear Reviewer,

We thank for the valuable feedback and for the opportunity to improve upon current limitations, which have been comprehensively addressed and implemented, as detailed below:

Comment 1: “Please provide the full form of PNA lectin. The abbreviation first appears in Section 2.1 line 123 without the full form.” 

Answer  1: We thank the reviewer for bringing this aspect to our attention. The full form of PNA lectin in the section 2.1 line 123.

Comment 2: “Authors should consider breaking Figure 3 into at least two different figures since it is too crowded in its current form. Text in Figure 3 (e) is hardly legible due to low resolution. Perhaps Figure 3 (b) – (d) that are showing results can be separated from (a) that is showing workflow.”

Answer 2: We thank the reviewer for the suggestion. The proposal to split Figure 3 into two figures has been well-received and implemented. Figure 3 now illustrates the workflow, while the new Figure 4 addresses the proteomics results. This adjustment has also improved the legibility of the text in Figure 3 (e).

Comment 3: Page 14 Lines 468-469 “After that, proteins were N-deglycosylated with PNGase F (1U/10μg) overnight at 37 °C” Please mention the number of hours for this incubation step. Overnight incubation is a vague term. Same for Trypsin digestion. Please provide the hours of incubation.”

Answer 3: We thank the reviewer for the comment. Overnight hours have been provided in the manuscript methods as requested.

Comment 4: “Please provide the volumes of 5 mM dithiothreitol and 10 mM iodoacetamide used for reduction, alkylation and quenching?”

Answer  4: We thank the reviewer for bringing this aspect to our attention. The volumes of 5 mM dithiothreitol and 10 mM iodoacetamide used for reduction, alkylation and quenching have been provided in the manuscript methods as request.

Comment 5: Provide the volume and amount of sample injected for LC-MS analysis. 

Answer  5: We would like to thank the reviewer for the comment. The volume and amount of sample injected for LC-MS analysis have been provided in the manuscript methods as request.